# Functional Magnetic Resonance Urography in Children—Tips and Pitfalls

**DOI:** 10.3390/diagnostics13101786

**Published:** 2023-05-18

**Authors:** Małgorzata Grzywińska, Dominik Świętoń, Agnieszka Sabisz, Maciej Piskunowicz

**Affiliations:** 1Applied Cognitive Neuroscience Lab., Department of Neurophysiology, Neuropsychology and Neuroinformatics, Medical University of Gdansk, 80-210 Gdansk, Poland; malgorzata.grzywinska@gumed.edu.pl; 22nd Department of Radiology, Medical University of Gdansk, 80-210 Gdansk, Poland; 31st Department of Radiology, Medical University of Gdansk, 80-210 Gdansk, Poland

**Keywords:** kidney perfusion, MR urography, urinary tract, MRI

## Abstract

MR urography can be an alternative to other imaging methods of the urinary tract in children. However, this examination may present technical problems influencing further results. Special attention must be paid to the parameters of dynamic sequences to obtain valuable data for further functional analysis. The analysis of methodology for renal function assessment using 3T magnetic resonance in children. A retrospective analysis of MR urography studies was performed in a group of 91 patients. Particular attention was paid to the acquisition parameters of the 3D-Thrive dynamic with contrast medium administration as a basic urography sequence. The authors have evaluated images qualitatively and compared contrast-to-noise ratio (CNR), curves smoothness, and quality of baseline (evaluation signal noise ratio) in every dynamic in each patient in every protocol used in our institution. Quality analysis of the image (ICC = 0.877, *p* < 0.001) was improved so that we have a statistically significant difference in image quality between protocols (χ^2^(3) = 20.134, *p* < 0.001). The results obtained for SNR in the medulla and cortex show that there was a statistically significant difference in SNR in the cortex (χ^2^(3) = 9.060, *p* = 0.029). Therefore, the obtained results show that with the newer protocol, we obtain lower values of standard deviation for TTP in the aorta (in ChopfMRU: first protocol SD = 14.560 vs. fourth protocol SD = 5.599; in IntelliSpace Portal: first protocol SD = 15.241 vs. fourth protocol SD = 5.506). Magnetic resonance urography is a promising technique with a few challenges that arise and need to be overcome. New technical opportunities should be introduced for everyday practice to improve MRU results.

## 1. Introduction

Imaging of the urinary system in children is mainly based on ultrasound (US), classical roentgenography, and scintigraphy. Each of these methods has its advantages as well as limitations [1,2,3]. Ultrasound is an optimal tool for visualizing kidneys, and their collecting systems are safe and commonly available. However, this method does not carry information about the renal function. Whereas classic roentgenography gives enough proper morphological evaluation of the collecting system and indirect information about the renal function. Unfortunately, in this method, we have a radiation burden, and unavoidable is the use of contrast media [4].

Another method used for imaging the urinary tract is computed tomography (CT). When the acquisition of images has become faster, this method has gained popularity; however, still limited due to radiation exposure and usage of an iodine contrast [5,6]. 

The gold standard in imaging renal function and determining split renal function is scintigraphic imaging with the use of a radionuclide Technetium-99m-mercaptoacetyltriglycine (99mTc-MAG3). This method, like the previous ones, has a few disadvantages—limitations in the detection of small subjects, low spatial and contrast resolution (inadequate anatomic information), usage of radionuclides, and hence ionizing radiation [7], which is why we are looking for an alternative.

Like Uro-CT, magnetic resonance urography (MRU) gives the possibility of assessing renal parenchyma, collecting systems, ureters, and bladder. The better signal, excellent contrast resolution, and lack of ionizing radiation make MRU a promising examination for non-invasive assessment of the urinary tract. However, it still requires a contrast medium, and it does not exceed Uro-CT in spatial resolution, as well as has a lower sensitivity for imaging renal stones [8,9,10]. 

Functional analysis of the MRU scan still requires external post-processing with relatively complicated software. It can be a limiting factor in the full routine implementation of functional analysis of the MRU, and the use of functional parameters of MRUs comparable to nuclear medicine [4,7,11,12,13,14]. 

MRU allows quantifying the perfusion of the parenchyma and glomerular filtration and visualizing of kidney excretory functions as well as urination [9,10,14,15,16]. All these advantages make MRU ideally suited for a comprehensive assessment of the upper urinary tract. However, the need for children’s anesthesia is still the main limitation [10,17,18].

For dynamic sequences, we use T1-weighted images followed by a paramagnetic intravenous contrast agent based on gadolinium. These sequences are usually preceded by diuretic injection approximately 15 min before contrast administration at a dose of 0.25 to 0.5 mg/kg body weight. The dynamic sequence is being performed in the coronal plane, which allows for assessing the anatomy of large kidney vessels, parenchyma, and collecting systems [4,17]. The imaging needs fat suppression to increase the visibility of the ureters. In this technique, the recommended (ESUR, Contrast Media Safety Guidelines 10.0) dose of gadolinium is 0.1 mmol/kg [19] (in our case: Gadobutrolum at flow rate of 0.5 mL/s). The use of diuretics is a supplement that can improve the excretion of the contrast agent and allows its greater dilution [20,21]. Contraindications for contrast agent use are: anuria, hypersensitivity to the drug, electrolyte imbalance, or hypotension and are cautious in patients allergic to sulphonamides. MRU technique can be performed in conjunction with a conventional MR for an integrated assessment of the urinary tract. 

However, pediatric MRU has some limitations: structures are much smaller than adult structures, heart rate, and respiratory frequency are higher than in adults, and patients are less likely to cooperate. Due to these adversities, acquisition parameters should be adjusted precisely to optimize the spatial and temporal resolution, and to make the acquisition time relatively short, but together with the right results. Therefore, imaging protocols and specific sequence parameters often need to adjust for the patients [4,22,23]. 

The aim of this study was the analysis of methodology for renal function assessment using 3T magnetic resonance imaging in children.

## 2. Materials and Methods

### 2.1. Subjects

A retrospective analysis of MRU 3D-Thrive dynamic sequence with contrast injection was performed on 91 patients (55 female, 36 male, mean age: 6.00 ± 5.84, range: 6 months–17 years).

The research was according to the Helsinki Declaration of 1975, as revised in 2000.

### 2.2. Imaging Protocol Evaluation

The examinations were performed using a Philips Achieva 3T TX magnetic resonance scanner (Philips Healthcare; Best, The Netherlands) with 16-channel coil dedicated to abdominal examinations. 

After taking the localizer sequence (a set of three-plane, low-resolution, large field-of-view images to localize part of the body to examination), morphological imaging sequences were acquired in three directions to provide anatomical orientation. The protocol included 3D thrive dynamic sequence for functional analysis. This sequence was changing over time; mainly, we changed parameters, such as matrix, the number of signals averaged (NSA), flip angle (FA), and number of dynamics. Every single examination was fitted for each patient to optimize time and spatial resolution—such as the field of view (FOV), and number of slices. Patients differ in age, weight, and height, which had an impact on the image signal and quality. Thus, we can put in approximate acquisition parameters (Table 1). 

For optimization time resolution and the acquisition matrix, we did use thinner slice, higher matrix, SENSitivity Encoding (SENSE), and recently we add ENCASE (Enhanced Coronal Acquisition with Sagittal Excitation [24]).

Both the protocol and the dynamic sequence were successively changed according to our knowledge, experience, and current technical possibilities (Table 1 and Table 2). The analysis was performed on images obtained in the dynamic schema with 3D-THRIVE sequence with the administration of the contrast agent. 

### 2.3. Assessment of Image Quality

Both kidneys were used for analysis. Image quality was performed on dynamic sequences before contrast by two examinators assessing in consensus. Visual assessment was performed by using scale of the visibility of corticomedullary differentiation in the time (1—poor, 2—moderate, 3—good, 4—excellent) [25,26]. The data were randomized for each author.

### 2.4. Contrast-to-Noise Ratio (CNR) and Signal-to-Noise-Ratio (SNR) Measurements

Contrast-to-noise ratio (CNR) and signal-to-noise-ratio (SNR) [27] is a measure used to determine image quality. This measurement was obtained with ROI (region of interest), which was placed in the cortex and medulla in the upper part of the kidney. ROI was as large as possible (Figure 1). The data were taken on the four places with the same size ROI in medulla and cortex and then averaged. Signal intensity on background noise was taken in the four ROIs in the background away from the body. CNR was calculated using the following formula [3,28]:$$CNR=\left|SNR_{cortex}-SNR_{medulla}\right|$$

In our case, when we use parallel image acquisition, SNR was defined as a relative mean signal in each subject divided by the standard deviation in the background.

### 2.5. Signal Intensity Curves Evaluation

Passing the contrast in time through specific anatomical parts of the kidneys can be represented by the changes in signal intensity curves depending on time from a specific ROI (Figure 1). This analysis was carried out in the Philips IntellinSpace Portal program. From the semiquantitative analysis, we can obtain information about relative enhancement, maximum enhancement, maximum relative enhancement, T0 (time of contrast in-flow), time to peak (time between the time of contrast inflow and time with a maximum of enhancement), wash in rate, washout rate, the brevity of enhancement, the area under the curve. However, for our needs, we used only T0 and TTP, to calculate RTT, and CTT in each kidney.

Functional analysis of kidneys was carried out in the program ChopfMRU (v 1.11, the Department of Radiology, at The Children’s Hospital of Philadelphia, https://www.parametricmri.com/ accessed on 17 May 2023) [9], which works on IDL Machine. Firstly, files of the MRU scan are exported into a single directory using free program DicomWorks^®^ (version 1.3.5, dicomworks.com accessed on 17 May 2023) [17] (Figure 2) and sorted.

The analysis in the ChopfMRU program has been divided into three stages:Separation of the aorta (Figure 3)—the number of time points was found so that the aorta was marked significantly against the background of the organs (the moment of the highest signal intensity in the vessel),Separation of the kidneys (Figure 4)—the number of time points was found where contrast is first seen in the calyces,Biophysical model analysis (Figure 5)—estimation of functional parameters for the aorta and each kidney.

From the biophysical model, the analysis included Patlak–Rutland method we can obtain information that was described by Khrichenko et al. [29]. For our usage, we focused on parameter: time to peak (TTP; time to achieve the maximum enhanced of the parenchyma; calculated automatically). 

The analysis of curves of signal intensity as a function of time from the ChopfMRU program was qualitatively compared by two authors with the curves obtained in the Philips IntelliSpace Portal program (Figure 6). The authors have evaluated qualitatively and compared contrast noise ratio (CNR), curves smoothness, and quality of baseline (evaluation signal noise ratio) in every dynamic in each patient in every protocol.

### 2.6. Statistical Analysis

The agreement between observers was measured by ICC (intraclass correlation coefficient). All results were tested for normal distribution with the Shapiro–Wilk test for each protocol. To improve the difference between data obtained with a different protocol, we used the Kruskal–Wallis H test and used U Manna–Whitney test to compare data between protocols, where Kruskal–Wallis H test shows a significant difference. The statistical analysis was prepared in IBM SPSS 25.0 (SPSS, Inc., Chicago, IL, USA).

## 3. Results

### 3.1. Evaluation of Imaging Techniques

In the first of three protocols, we used dynamic sequences of the angle of the oblique-coronal plane. At the fourth, the maximum angle of the plane can be 5 degrees, because we use ENCASE and for this angulation from the coronal orientation is limited to +/−5 degrees for all directions. However, we used this modality mainly to suppress the breathing artifacts.

We placed the patient with his or her hands up; this allowed us to adjust the kidney’s long axis, thus compensating for the restriction—the second problem which we have a problem with analysis enhanced plot. We increased the number and frequency of consecutive dynamics in the initial phase. For the first protocol, we had 30 s break between dynamics, while in the fourth protocol, we use for beginning, dynamics were acquired one by one, without interruption. Then a few minutes of scanning—30 s break. Thanks to that, we have more information about kidney perfusion from the first minutes after contrast injection until notice a contrast in the ureter below the lower pole of the kidney.

### 3.2. Assessment of Image Quality

The agreement between observers was 0.877 (a high intraclass correlation coefficient) with *p* < 0.001. Therefore, the results in Table 3 present descriptive statistics of data collected by one observer. The results for each protocol were tested for normal distribution with the Shapiro–Wilk test. This test showed that obtained data do not have a normal distribution (*p* < 0.05). Hence, gain data were tested with the Kruskal–Wallis H test. This test shows that there was a statistically significant difference in image quality between protocols (χ^2^(3) = 20.134, *p* < 0.001). To improve which group of the protocols is a significant difference (Figure 7), we used the U Manna–Whitney test. We obtained that the significant difference was between the first and fourth protocol (Z = −3.423, *p* = 0.001), the second and third (Z = −2.323, *p* = 0.020), second and fourth (Z = −4.212, *p* < 0.001).

### 3.3. Contrast-to-Noise Ratio (CNR) and Signal-to-Noise-Ratio (SNR) Measurements

Results of signal-to-noise-ratio (SNR) and contrast-to-noise-ratio (CNR) for each protocol did not have a normal distribution (*p* < 0.05). The mean value of CNR was almost the same for every protocol (Table 4) (Figure 8, Figure 9 and Figure 10). However, the median in the fourth protocol had the highest value, and this may indicate a better contrast of the image in this protocol. By analyzing the results obtained for SNR in medulla and cortex (Table 5), we can expect significant differences between the older protocols and the latest ones.

Therefore, obtained data were tested with the Kruskal–Wallis H test. This test shows that there was a statistically significant difference in SNR in the cortex (χ^2^(3) = 9.060, *p* = 0.029) and medulla (χ^2^(3) = 8.114, *p* = 0.44) between protocol, but in CNR we do not obtain the significant difference between protocols (χ^2^(3) = 4.542, *p* = 0.209). 

To improve which group of the protocol is a significant difference in SNR, we used the U Manna–Whitney test. We obtained that the significant difference was in the:Cortex: 2nd and 3rd (Z = −2.429, *p* = 0.015), 2nd and 4th (Z = −2.626, *p* = 0.009),Medulla: 2nd and 3rd (Z = −2.324, *p* = 0.020), 2nd and 4th (Z = −2.626, *p* = 0.009).

### 3.4. Quality of Enhancement Curves

The enhancement curve of dynamic sequences for each patient was assessed. Changing the parameters of acquisitions of new dynamics during the initial phase until excretion, increased the effectiveness of the renal perfusion assessment. 

The use of techniques shortening the duration of one dynamic, and techniques changing the direction of data collection (ENCASE), has improved the quality of the data obtained. There are no graph line jumps (Figure 11 and Figure 12a) at subsequent time points on the obtained signal intensity curves which are smoother and out of breathing artifacts (Figure 6 and Figure 12a). That illustrates how much information we lost in the first protocol, and how much information we can obtain with the fourth protocol. To improve this, we check TTP in the aorta (the time between highest signal intensity and point before contrast injection) in IntelliSpace Portal and ChopfMRU, and we checked TTP standard deviation for each protocol (Table 6). We used data in the aorta because we have a constant contrast injection flow and time-to-peak in the aorta should be approximately comparable in each patient without the difference in weight, age, height, or kidney disease. The obtained results show that with the newer protocol, we obtain lower values of standard deviation for TTP in the aorta (in ChopfMRU: first protocol SD = 14.560 vs. fourth protocol SD = 5.599; in IntelliSpace Portal: first protocol SD = 15.241 vs. fourth protocol SD = 5.506). This may indicate that with the latest protocol, we can achieve repeatable results with a possible low data loss.

## 4. Discussion

The present study evaluated imaging techniques for assessing kidney perfusion. The fourth protocol significantly improved image quality and renal perfusion assessment by suppressing breathing artifacts and increasing the number and frequency of consecutive dynamics during the initial phase until excretion. These findings have important clinical implications, as imaging techniques are critical for diagnosing and managing kidney disease.

The most significant technical problem concerning functional MRU imaging is obtaining the balance between spatial resolution, time resolution, and adjusting the acquisition matrix to the patient’s age [30]. Additionally, the qualitative analysis has problems with ROI (region of interest) positioning and respiratory artifacts. To address these limitations, techniques such as ENCASE and SENSE, and increasing the matrix size adequately to the patient’s size must be implemented [31,32].

To improve time resolution, we started using compressed sensing, a method for accelerated MR data acquisition based on the semi-random sampling of k-space. Adding compressed SENSE to multiple sequences in an exam can accelerate our examinations by 20–40%, which is very good for the patients and examination quality [33,34]. This is obvious because the shorter the examination time, the greater the patient’s comfort and, thus, the fewer motion artifacts adversely affecting the images. The other option to improve the quality of obtaining images could be a 4D MRI—RT respiratory self-gating sequence. This technique captions organ motion under free-breathing and automatic sorting obtaining data without external respiratory devices [35].

The proposed fourth protocol overcomes these mentioned above limitations by the highest matrix (384/384), the shortest time of the single dynamic sequence (8 s), and the highest number of dynamic sequences (40–50).

In the presented study, the analysis of the resolution based on corticomedullary differentiation showed a significant difference between the fourth and first protocol (Z = −3.423, *p* = 0.001), as well as between the second and third (Z = −2.323, *p* = 0.020), and second and fourth (Z = −4.212, *p* < 0.001) protocols. This evaluation is because diagnosis is based primarily on a visual assessment of the dynamics, and the temporal resolution greatly influences the quantitative evaluation of contrast intensity curves during postprocessing. Results obtained from the analysis of TTP in the aorta showed that with the latest, updated protocols, lower values of standard deviation were characterized, which may indicate more repeatable results (in ChopfMRU: first protocol SD = 14.560 vs. fourth protocol SD = 5.599; in IntelliSpace Portal: first protocol SD = 15.241 vs. fourth protocol SD = 5.506). In addition, SNR significantly differs in the medulla and cortex between the second and third (cortex: Z = −2.429, *p* = 0.015; medulla: Z = −2.324, *p* = 0.020), and the second and fourth (cortex: Z = −2.626, *p* = 0.009; medulla: Z = −2.626, *p* = 0.009) protocols. When we achieve optimal time resolution, renal function can be assessed more accurately. Based on our results’ lowest standard deviation (TTP in the aorta), we can assume that the fourth protocol can produce repeatable results with minimal data loss. All the changes introduced have led to better visibility of corticomedullary differentiation, as confirmed by visual and SNR evaluations. This improvement will help diagnosis, which is primarily based on qualitative evaluation. Obtained results illustrate that our changes helped us obtain higher-resolution images without compromising time resolution.

Moreover, the changes introduced in the proposed protocol allowed for a more accurate assessment of renal status, and the precise quantitative and qualitative analysis and calculations of the values of signal intensity curves in individual segments and parts of the kidneys were improved.

While the present study focused on MRI techniques, it is worth noting that other imaging modalities, such as CT and ultrasound, are also commonly used to diagnose and manage kidney disease [3,14,36]. The combination of fMRI with contrast-enhanced ultrasound (CEUS) may be particularly interesting. Hence, future studies could compare the efficacy of these imaging modalities to determine the most effective and appropriate technique for different clinical scenarios.

The findings of this study have potential implications for future research in the field of renal imaging. For instance, the fourth protocol presented in this study could be further optimized and improved upon in future studies using artificial intelligence (AI) and deep learning techniques [1].

It is important to note that the present study has some limitations. For instance, the study was limited to a small sample size, and the results may not be generalizable to larger populations or different clinical scenarios. Additionally, the study only evaluated one aspect of renal imaging, and future studies could explore other aspects of renal function imaging supported with AI [37,38,39,40].

In summary, there are several methods to improve the quality of the resulting images, including:High-field MRI: using a higher magnetic field strength can improve the signal-to-noise ratio and overall image quality.Parallel imaging: this technique allows data to be acquired from multiple coils simultaneously, reducing acquisition time and improving spatial resolution.Motion correction: motion artifacts can negatively impact image quality, but motion compensation algorithms can compensate for patient movement and improve image quality.Compressed sensing: this signal processing technique can reconstruct high-quality images from undersampled data.Deep learning: the newest technique uses deep learning methods such as convolutional neural networks to improve image quality by reducing noise and enhancing contrast.

## 5. Conclusions

Functional magnetic resonance imaging (fMRU) is a non-invasive imaging modality that combines functional MRI and urography to provide a detailed picture of the functioning urinary tract. It is a promising technique with a few challenges that arise and need to be overcome. New technical opportunities should be introduced for everyday practice to improve MRU results.

## Figures and Tables

**Figure 1 diagnostics-13-01786-f001:**
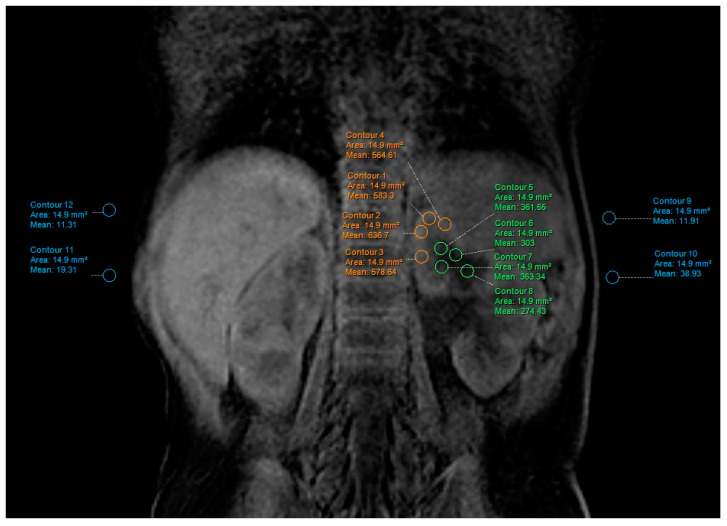
Signal noise ratio measurements.

**Figure 2 diagnostics-13-01786-f002:**
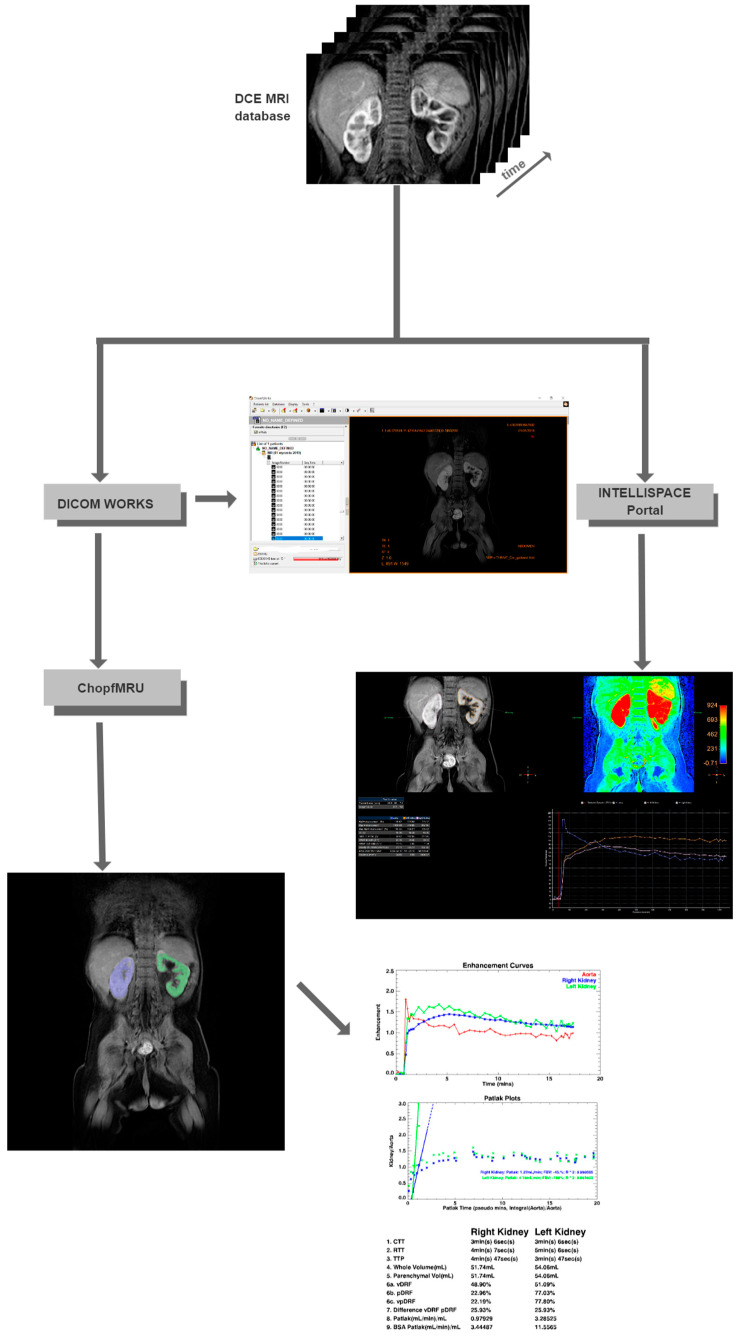
Model of the process of analysis.

**Figure 3 diagnostics-13-01786-f003:**
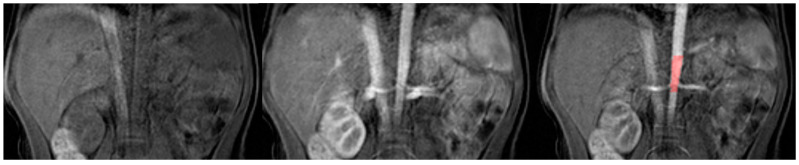
Example of separation of the aorta.

**Figure 4 diagnostics-13-01786-f004:**
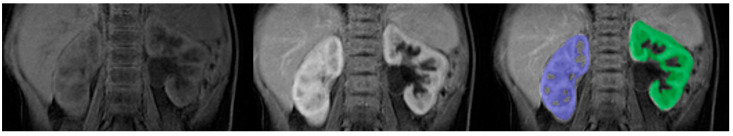
Example of separation of kidney.

**Figure 5 diagnostics-13-01786-f005:**
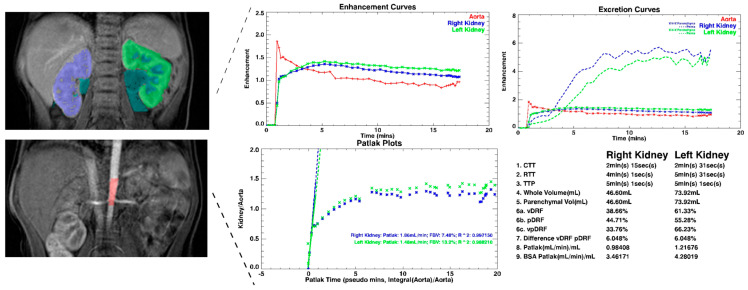
Example of mathematical analysis.

**Figure 6 diagnostics-13-01786-f006:**
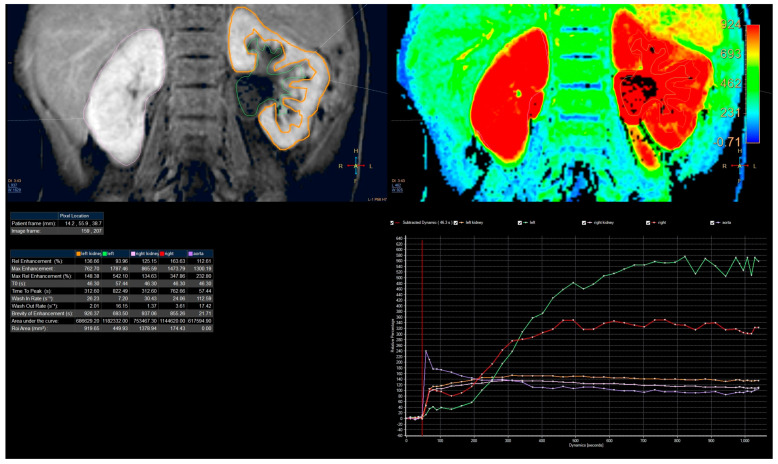
Example of analysis in the IntelliSpace Portal program. Example of analysis curves of 4th protocol obtained on IntelliSpace Portal 10.0.

**Figure 7 diagnostics-13-01786-f007:**
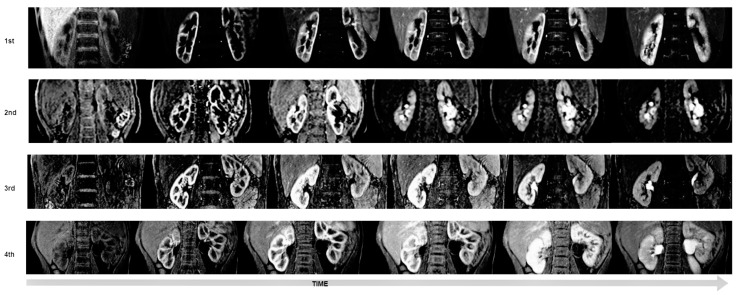
Examples of evaluation of quality on the dynamic sequence in every protocol, visualization in time after contrast was injected.

**Figure 8 diagnostics-13-01786-f008:**
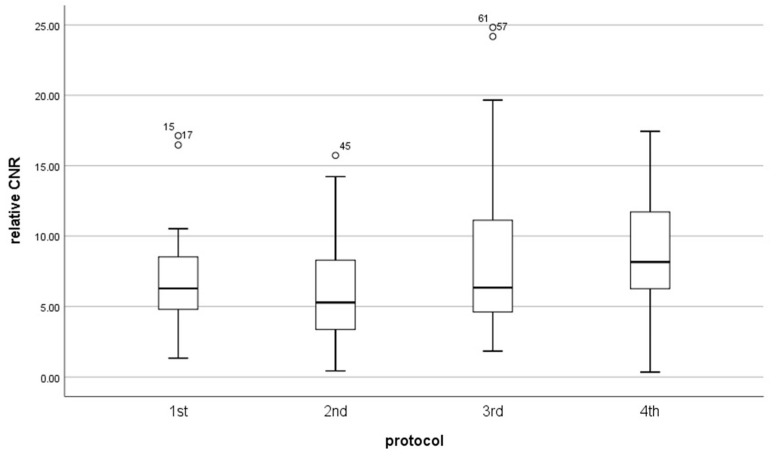
Box plot illustrates the changing of CNR for each protocol.

**Figure 9 diagnostics-13-01786-f009:**
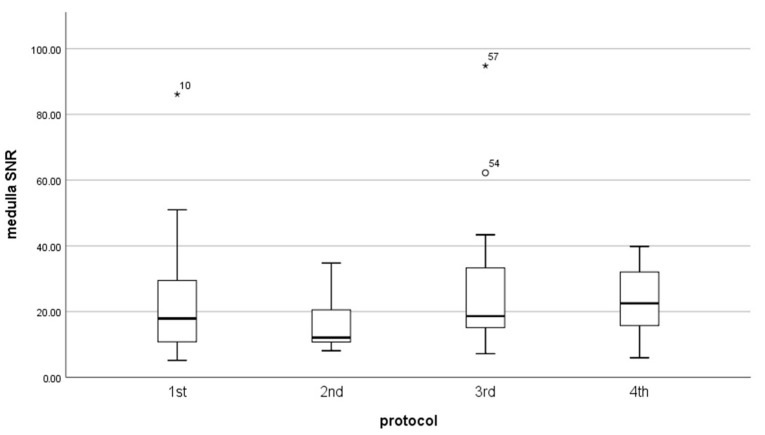
Box plot illustrates the changing of SNR in medulla for each protocol.

**Figure 10 diagnostics-13-01786-f010:**
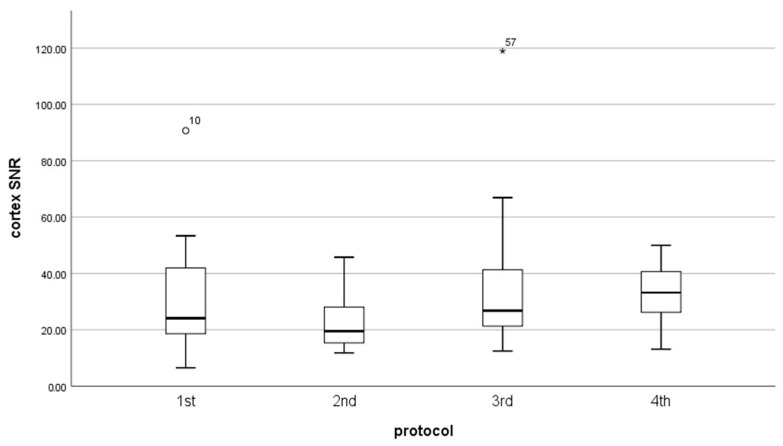
Box plot illustrates the changing of SNR in cortex for each protocol.

**Figure 11 diagnostics-13-01786-f011:**
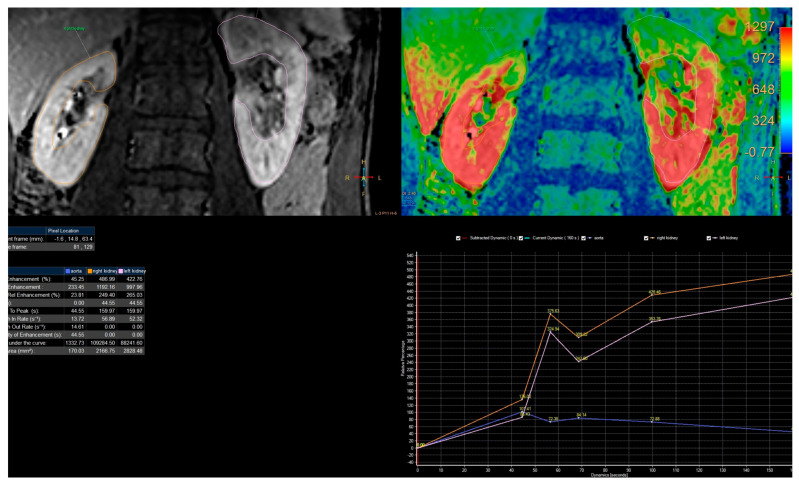
Example of analysis curves of 1st protocol obtained on IntelliSpace Portal 10.0.

**Figure 12 diagnostics-13-01786-f012:**
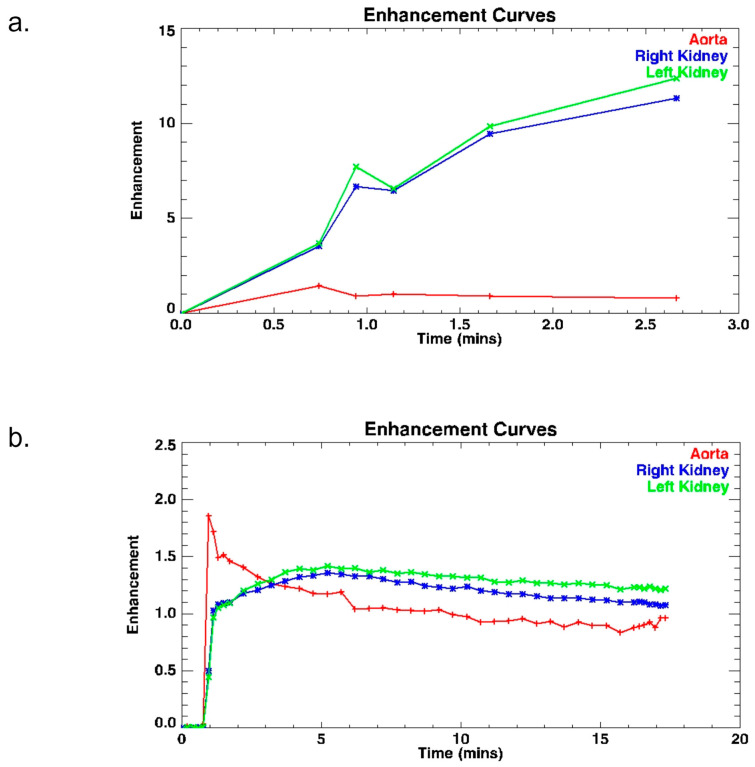
Example to obtaining enhancement curves (**a**) 1st protocol, (**b**) 4th protocol; obtained on Chop fMRU program.

**Table 1 diagnostics-13-01786-t001:** Evaluation of examination protocol.

1st Protocol		2nd Protocol		3rd Protocol		4th Protocol	
1	Survey	1	Survey	1	Survey	1	Survey
2	T1_TFE_IP_Cor_FB	2	T1_TFE_IP_Cor_FB	2	mDixon_Tra	2	T2W_TSE_Tra_HR
3	T1_TFE_IP_Tra_FB	3	VISTA_COR	3	VISTA_COR_Sense	3	VISTA_COR_Sense
4	BTFE_SPIR_COR_FB	4	BTFE_SPIR_SAG_FB	4	DWI_5b_Tra_navi	4 ^1^	BTFE_SPIR_SAG_FB
5	T2_TSE_TRA_FB	5	T2_TSE_HR_TRA_FB	5	T2W_TSE_Tra_HR	5	mDixon_Tra
6	T2W_SPAIR_TRA_FB	6	STIR_Tra_FB	6	T2_SPAIR_TRA	6	DWI_5b_Tra_navi
7	VISTA_COR	7	DWI_5b_Tra_navi	7	BTFE_SPIR_SAG_FB	7	STIR_Tra_FB
8	sMRCP_3D_HR_COR	8	mDIXON_ TRA_FB	8	THRIVE_COR_3D	8 ^2^	sMRCP_3D_HR_COR
9	e-THRIVE_COR_FB	9	THRIVE_COR_FB	9	sT1W_FFE_IP	9	e-THRIVE_COR_FB
10	T1_TFE_IP_COR_FB	10	T1_TFE_IP_COR_FB			10	mDixon_Tra

TFE—turbo field echo; IP—in phase; FB—free breath, Navi—breathing navigator, TSE—turbo spin echo; SPAIR—spectral attenuated inversion recovery; VISTA—Volume isotropic turbo spin-echo acquisition = 3D-FAST SPIN ECHO; MRCP—magnetic resonance cholangiopancreatography; mDixon—time-consuming acquisition of in-phase and opposed-phase gradient-echo images; eTHRIVE—enhanced T1 high-resolution isotropic volume excitation; T1—the time constant for regrowth of longitudinal magnetization; T2—is the time constant for decay/dephasing of transverse magnetization; DWI—diffusion-weighted imaging. ^1^ This sequence is useful, when patient have uretero-pelvic junction structure obstruction. ^2^ This sequence is useful, when VISTA sequence have not enough good quality.

**Table 2 diagnostics-13-01786-t002:** Acquisition parameters of the dynamic sequence.

Parameters	1st Protocol	2nd Protocol	3rd Protocol	4th Protocol
Slice thickness/gap (mm)	5	3	3	4
Suppress fat	SPAIR	SPIR	SPIR	SPAIR
Flip angle	10	25	25	10
Number of averages	3	1	1	1
TE/TR (ms)	Default	Default	default	default
Matrix	≈214/214	≈214/214	≈232/232	≈384/384
Time (s) per dynamic	≈10	≈12	≈8	≈8
Breath hold	Free breath	Free breath	Free breath	Free breath
Sense	-	+	+	+
ENCASE	-	-	-	+
Number of dynamics	10	25	≈25–30	≈40–50
Number of dynamics without contrast	1	1	5–7	>7
Delay between dynamics	30 s	30 s	30 s	For the beginning, dynamics were acquired one by one, without delay. After a few minutes—30 s
Plane	Coronal with the angle of the oblique-coronal plane in long axis kidney	Coronal with the angle of the oblique-coronal plane in long axis kidney	Coronal with the angle of the oblique-coronal plane in long axis kidney	Coronal with a maximum of 5 degrees

**Table 3 diagnostics-13-01786-t003:** Results of visual assessment with a 4-point scale.

Number of Protocol	Visual Assessment with a 4-Point Scale					
	N	Minimum	Maximum	Mean	Median	Standard Error
1st protocol	20	2.00	4.00	2.85	3	0.18
2nd protocol	32	1.00	4.00	2.78	3	0.13
3rd protocol	22	1.00	4.00	3.27	3	0.19
4th protocol	17	3.00	4.00	3.77	4	0.11

Where: N—the number of patients in the group, minimum—minimum value of results, maximum—maximum value of results, median—optimal prediction of values using a single number.

**Table 4 diagnostics-13-01786-t004:** Results of contrast-to-noise ratio (CNR).

Number of Protocol	Contrast to Noise Ratio (CNR)					
	N	Minimum	Maximum	Mean	Median	Standard Error
1st protocol	20	1.33	17.13	7.07	6,28	0.91
2nd protocol	32	0.43	15.73	6.15	5.29	0.72
3rd protocol	22	1.83	24.82	8.84	6.34	1.43
4th protocol	17	0.35	17.44	8.59	8.16	1.04

Where: N—the number of patients in the group, minimum—minimum value of results, maximum—maximum value of results, median—optimal prediction of values using a single number.

**Table 5 diagnostics-13-01786-t005:** Results of signal-to-noise ratio (SNR) in medulla and cortex.

Number of Protocol	Signal to Noise Ratio (SNR) in Medulla						Signal to Noise Ratio (SNR) in Cortex				
	N	Minimum	Maximum	Mean	Median	Standard Error	Minimum	Maximum	Mean	Median	Standard Error
1st protocol	20	5.18	86.11	23.867	17.89	4.39	6.51	90.68	30.98	24.10	4.24
2nd protocol	32	8.09	34.75	16.22	12.09	1.31	11.82	45.75	22.38	19.55	1.61
3rd protocol	22	7.20	94.76	26.37	18.62	4.24	12.48	118.94	35.22	26.80	5.07
4th protocol	17	5.95	39.81	23.14	22.52	2.38	13.12	49.99	31.74	33.21	2.91

Where: N—the number of patients in the group, minimum—minimum value of results, maximum—maximum value of results, median—optimal prediction of values using a single number.

**Table 6 diagnostics-13-01786-t006:** Standard deviation obtained on TTP in the aorta.

	TTP Standard Deviation	
	IntelliSpace Portal	ChopfMRU
1st protocol	15.241	14.560
2nd protocol	10.550	13.768
3rd protocol	9.214	11.520
4th protocol	5.506	5.599

## Data Availability

The data presented in this study are available on reasonable and qualified research request from the corresponding author. Data requestors will need to sign a data access agreement.

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
