# Peer review of "Functional Magnetic Resonance Urography in Children—Tips and Pitfalls"

_diagnostics, 2023, doi:10.3390/diagnostics13101786_

Round 1

Reviewer 1 Report

I read with interest this interesting and nicely written manuscript. I believe that the study is well conducted and especially is well described in methods. Results are deeply presented, and tables / figures are of help. I sincerely would implement the discussion session, which is too short. Similarly, in the discussion you acknowledge the technical limitations encountered during the study period but you don’t declare the study / manuscript limitations. Please expand this session.

Author Response

Thank you for reading our manuscript and for your positive feedback. We appreciate your comments on the description of the method and the presentation of the results. We are glad that you found the manuscript interesting and well-written.

Suggestions for expanding the area of ​​discussion are on point. We agree that this section is more comprehensive and provides a detailed analysis of our findings and their implications. Your feedback is valuable to us, and we appreciate your thoughtful comments. We will consider your suggestions when revising the manuscript and strive to provide a more comprehensive and informative discussion section that addresses the study's limitations.

Reviewer 2 Report

This is an interesting original article on a current topic. It would be necessery to expand the Discussion section by comparing results with the findings of other studies.

Author Response

Thank you for reading our article and for this valuable feedback. We appreciate your interest in our study and your suggestion to expand the discussion section by comparing our results with other studies.

We agree that comparing our findings with those of other studies provides valuable context and helps readers better understand the implications of our findings. We will consider this feedback when revising the manuscript and how to incorporate this additional information into the discussion section.

Round 2

Reviewer 2 Report

Well-corrected manuscript.